# Computerized Adaptive Testing via Collaborative Ranking

**Zirui Liu**[1], **Yan Zhuang**[1], **Qi Liu**[1,2]*, **Jiatong Li**[1], **Yuren Zhang**[1], **Zhenya Huang**[1],
**Jinze Wu**[3], **Shijin Wang**[3]

1: State Key Laboratory of Cognitive Intelligence,
University of Science and Technology of China
2: Institute of Artificial Intelligence, Hefei Comprehensive National Science Center
3: iFLYTEK Co., Ltd
{liuzirui,zykb,cslijt,yr160698,hxwjz}@mail.ustc.edu.cn
{qiliuql,huangzhy}@ustc.edu.cn, sjwang3@iflytek.com

## Abstract

As the deep integration of machine learning and intelligent education, Computerized Adaptive Testing (CAT) has received more and more research attention. Compared to traditional paper-and-pencil tests, CAT can deliver both personalized and interactive assessments by automatically adjusting testing questions according to the performance of students during the test process. Therefore, CAT has been recognized as an efficient testing methodology capable of accurately estimating a student's ability with a minimal number of questions, leading to its widespread adoption in mainstream selective exams such as the GMAT and GRE. However, just improving the accuracy of ability estimation is far from satisfactory in the real-world scenarios, since an accurate ranking of students is usually more important (e.g., in high-stakes exams). Considering the shortage of existing CAT solutions in student ranking, this paper emphasizes the importance of aligning test outcomes (student ranks) with the true underlying abilities of students. Along this line, different from the conventional independent testing paradigm among students, we propose a novel collaborative framework, Collaborative Computerized Adaptive Testing (CCAT), that leverages inter-student information to enhance student ranking. By using collaborative students as anchors to assist in ranking test-takers, CCAT can give both theoretical guarantees and experimental validation for ensuring ranking consistency.

## 1 Introduction

With the rapid advancements in computer science, online education has undergone significant transformation, reshaping and displacing traditional offline educational assessment techniques. In this evolving landscape, Computerized Adaptive Testing (CAT) [1, 2] has emerged as a prominent methodology for standardized testing, widely adopted in selective exams such as the GMAT [3], GRE [4], and TOEFL [5]. Diverging from traditional paper-and-pencil tests, CAT offers personalized and interactive assessments, where the difficulty and characteristics of questions are continuously adapted based on real-time responses. By aligning questions with current estimation of students' abilities, CAT refines the estimation process each iterative step [6]. Upon test completion, the final ability score shown in Figure 1(a) is provided as score report to students. This score plays a pivotal role in influencing their educational and career prospects.

---

*Corresponding Author.

38th Conference on Neural Information Processing Systems (NeurIPS 2024).

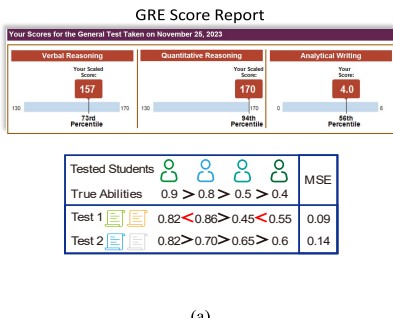
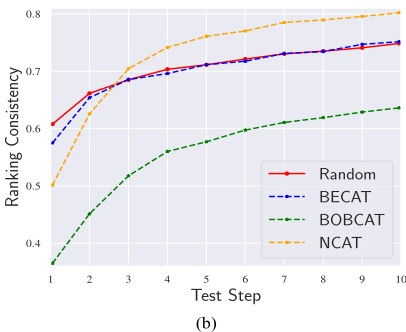

(a)                      (b)

Figure 1: (a) The score report provided by GRE and an example to show that a low MSE cannot guarantee the correct ranking of students' testing results. (b) This line chart shows the performance of previous CAT methods in ranking, and it can be seen that the method that performs state-of-the-art (BECAT) in accuracy may only achieve the effect of random selection in ranking.

However, while massive efforts have been made on optimizing the accuracy of ability estimation via improvements to the question selection algorithms [7, 8, 9, 10, 11, 12], it is crucial to underscore that *accurate ability estimation does not inherently guarantee correct student ranking*. As illustrated in Figure 1(a), minimizing mean squared error (MSE) in ability scores does not always translate into accurate rankings of students. In fact, even state-of-the-art (SOTA) question selection algorithms with superior accuracy performance can exhibit inconsistencies in ranking performance, sometimes performing worse than random selection methods, as presented in Figure 1(b). Meanwhile, the asynchronicity and independency between different students in the CAT test process [13, 14] is a significant technical challenge in achieving accurate ability ranking. This issue prevents the utilization of all students' testing information together for question selection to enhance ranking precision among students, thereby complicates the resolution of the ranking consistency issue in CAT.

To address this challenge, we propose a novel framework—Collaborative Computerized Adaptive Testing (CCAT), which introduces a collaborative learning [15, 16] approach that leverages data from collaborative students as ranking anchors. This framework facilitates interaction among test-takers, allowing for more robust ranking results. Importantly, we also present a theoretical analysis that demonstrates how, with a sufficient number of collaborative students, the ranking consistency error can be significantly reduced to an acceptable level. In summary, our contributions are:

- To the best of our knowledge, this is the first research to unveil the ranking consistency dilemma inherent in CAT, by providing its formal definition and approximation. This discovery has enabled us to significantly refine the objectives of CAT, which is a vital advancement for its deployment in high-stakes examination contexts.

- We introduce a novel, collaboration-based methodology that enhances both question selection and ability estimation to minimize ranking inconsistency, providing theoretical guarantees for ranking consistency even with a limited number of questions.

- Our methodology is general enough to integrate with existing question selection algorithms. Empirical results on extensive real-world educational datasets proves the effectiveness of CCAT, manifesting in an average 5% rise in ranking consistency compared with other methods, and this improvement is more significant in the short test scenarios.

## 2 Related Work

CAT is designed to efficiently and accurately estimate students' abilities[2]. It is widely employed in various competitive exams, including the GRE. CAT essentially operates in two stages: first, it uses methods such as Item Response Theory (IRT) [17] to estimate students' abilities. Subsequently, it uses these estimations to select the next question for each student. The following paragraphs separately outline Item Response Theory and common question selection algorithms used in CAT.

**Item Response Theory (IRT).** IRT is a psychological measurement theory predominantly employed in education to estimate students' abilities [17, 18, 19]. It posits that examinees' abilities remain constant throughout a test, and their performance depends solely on their ability and the information provided by the questions . The standard model is the two-parameter logistic (2PL) model: $P_j(\theta) = P(y_j = 1) = \sigma(a_j(\theta - b_j))$, where $\sigma(x) = \frac{1}{1+e^{-x}}$ is sigmoid function and $y_j = 1$ indicates a correct response to question $j$. The parameters $a_j, b_j \in \mathbb{R}$ represent the discrimination and difficulty of question $j$. These parameters are estimated by algorithms such as Markov Chain Monte Carlo (MCMC) [20, 21] and Gradient Descent (GD) [22, 23] before testing. $\theta \in \mathbb{R}$ represents the student's ability, which is estimated using the maximum likelihood method at each step $t$:

$$\theta^t = \arg\max_{\theta \in \Theta} \sum_{j=1}^{t} y_j \ln P_j(\theta) + (1 - y_j) \ln (1 - P_j(\theta)). \tag{1}$$

In recent years, the increasing studies [24, 25, 26, 27, 28] leveraging the rapid advancements in deep learning technologies (e.g., the neural networks) have significantly enhanced the accuracy of student ability estimation. For example, NeuralCD [24] leverages a non-negative fully connected neural network to capture the complex student-question interactions to achieve a more accurate estimation.

**Selection Algorithms.** Research on selection algorithms can be categorized into two main approaches: traditional rule-based algorithms and data-driven algorithms. Firstly, traditional question selection algorithms[29, 30, 31] view CAT as a parameter estimation problem. They calculate the information value of each question based on the student's current proficiency and select the question with the maximum information value[32], typically using metrics such as Fisher Information (FSI) [32] and Kullback-Leibler Information (KLI) [33]. Subsequently, in order to optimize the accuracy of the test result directly, researchers have proposed methods such as MAAT [34], BOBCAT [35] and NCAT [36], which are based on active learning [37], meta-learning [38] and reinforcement learning [39]. Recently, BECAT [40] proposes to use the ability estimated by student's full responses on the entire question bank as the true value and solve the CAT problem using a data efficiency method [41].

In fact, in many exams, especially selective exams, the ranking of grades is usually one of the most important bases for employment. So we argue that the requirement of students in CAT is not necessarily a more precise estimation of their abilities on the test set. Rather, CAT should ensure that students with stronger abilities receive better rankings. Consequently, we establish the ranking consistency of CAT as our primary objective.

## 3 Ranking Consistency of CAT

We first assume that the testing step in CAT is uniformly $T$ steps and all the selected questions come from question bank $Q$. The questions answered by each student constitute a subset $S \subseteq Q$. For each step $t$, the student's ability estimated by IRT is $\theta^t$ and the student's final result is $\theta^T$ when the test stops. For traditional CAT methods, the goal is that test results $\theta^T$ should be as close as possible to the true abilities of students $\theta^*$ with fewer questions [40, 42]:

$$\min_{|S|=T} ||\theta^T - \theta^*||, \tag{2}$$

where $\theta^*$ is approximated by the abilities of students estimated by their full responses to the entire question bank $Q$ [40]. However, as previously mentioned, CAT often prioritize the issue of ranking among students over merely improving the accuracy of $\theta^T$. For instance, if students learn that a peer with lower true ability outperforms them in CAT, they may question the fairness of the exam [43]. Thus, we define the consistency of CAT ranking as follows:

**Definition 1.** (Ranking Consistency of CAT) In computerized adaptive testing, the true abilities of two students are represented by $\theta_1^*$ and $\theta_2^*$. The testing results of these two students on subsets $S_1$ and $S_2$ of question bank $Q$ are denoted by $\theta_1^T$ and $\theta_2^T$. The ranking consistency of testing demands that students with higher true abilities should also exhibit higher testing results:

$$\max_{|S_1|=|S_2|=T} P(\theta_1^T > \theta_2^T | \theta_1^* > \theta_2^*). \tag{3}$$

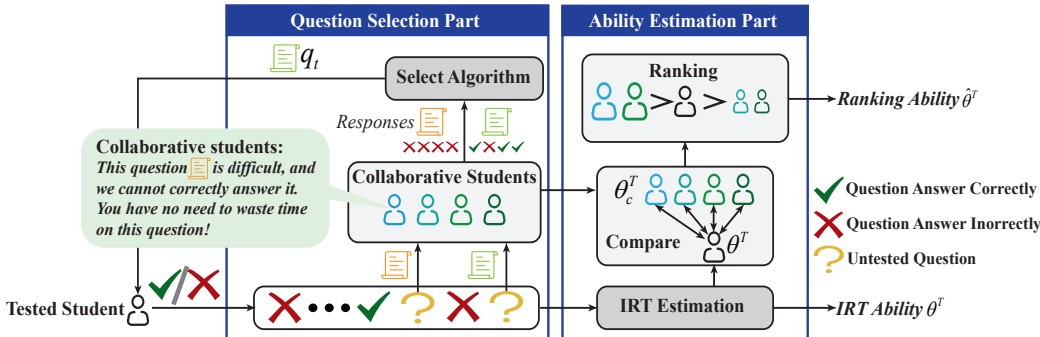

Figure 2: The structure of CCAT framework. CCAT consists of two parts: question selection and ability estimation. The question selection part utilizes the performance of collaborative students in answering various questions to select appropriate questions for the tested student, and the ability estimation part ranks the tested student with collaborative students and uses the ranking as the test result.

Given the varied performance, queries, and progress of the students undergoing testing, they remain independent during the CAT process. Consequently, it is impractical to intervene in ranking consistency by selecting questions based on each others' performance in the test. This complicates the direct optimization of this problem.

## 4 The CCAT Framework

To address the problem of ranking consistency, in this section, we first introduce the concept of collaborative students as anchors for the tested students. Then we elucidate their application in question selection and ability estimation. Finally, we conducted a theoretical analysis of the collaborative student method, demonstrating that while the ranking of the tested students among collaborative students may not be entirely accurate, the likelihood of achieving ranking consistency in CAT can reach at least $1 - \delta$ when a sufficient number of collaborative students are available.

**Definition 2.** (Collaborative Students) Collaborative students represent a group with $M$ students who are utilized as anchors to assist in ranking test-takers [44, 45]. It can be assumed that collaborative students have already completed answering all questions in the question bank $Q$, and their abilities on question bank $Q$ or subset $S(|S| = T)$ are $\theta_c^*$ and $\theta_c^T$, which can be obtained easily.

Due to the absence of information disclosure between any two students during the testing process, we cannot directly intervene in their ranking relationship. Nonetheless, since the collaborative students answered every question from the question bank, we can hypothesize that each collaborative student will accompany the tested students in responding to the same questions during the test. This could facilitate the establishment of relationships among the tested students.

Specifically, when two students, $A$ and $B$, answer distinct sets of questions, say $q_1, q_2, q_3$ for student $A$ and $q_4, q_5, q_6$ for student $B$, inconsistencies may arise due to the dissimilarity of the questions. However, each collaborative student can compare their performance with both students $A$ and $B$. For instance, a collaborative student can assess her performance on questions $q_1, q_2, q_3$ alongside student $A$ and on questions $q_4, q_5, q_6$ alongside student $B$. If the collaborative student finds that her abilities exceed those of student $A$ but fall short of student $B$, she will provide valuable information for accurately ranking students $A$ and $B$.

### 4.1 Problem Approximation

As previously mentioned, our goal is to establish the ranking relationship between tested students by comparing with collaborative students. Obviously, the first step in ensuring the ranking consistency among tested students is to establish ranking consistency between the collaborative students and the tested students:

$$\max_{|S|=T} P(\theta^T > \theta_c^T | \theta^* > \theta_c^*, S). \tag{4}$$

In Section 2, we outlined the estimation method for $\theta$ in Item Response Theory, as presented in Equation (1). Utilizing this formula, we can derive the subsequent lemma, which aids in simplifying the optimization objective.

**Lemma 1.** *Given two students, whose responses on $S(|S| = T)$ are $y_1, y_2, \cdots, y_T$ and $\tilde{y}_1, \tilde{y}_2, \cdots, \tilde{y}_T$, their testing abilities on $S$ are $\theta^T, \tilde{\theta}^T$, which are estimated by IRT with parameters $a_i, b_i$. We can prove that if $(\theta^T - \tilde{\theta}^T) > 0$, then $\sum_{i=1}^{T} a_i(y_i - \tilde{y}_i) > 0$, vice versa.*

Lemma 1 posits that if two students are tested on the same question subset, the term $\sum_{i=1}^{T} a_i(y_i - y_i^c)$ can be used to replace $\theta^T - \theta_c^T$ because they share the same sign (either positive or negative). This substitution leads to a more streamlined formulation of the objective:

$$
\begin{aligned}
P(\theta^T > \theta_c^T | \theta^* > \theta_c^*, S) = P & \left( \sum_{q_i \in S} a_i(y_i - y_i^c) > 0 | S, \theta^* > \theta_c^* \right) \\
\Rightarrow & \sum_{q_i \in S} a_i P(y_i > y_i^c | \theta^* > \theta_c^*) \\
= & \sum_{q_i \in S} R(q_i | \theta^* > \theta_c^*),
\end{aligned}
\tag{5}
$$

where $R(q_i | \theta^* > \theta_c^*) = a_i P(y_i = 1 | \theta^*) P(y_i^c = 0 | \theta^* > \theta_c^*)$, $y_j^c$ and $y_j$ represent the responses of collaborative students and tested students to question $j$ respectively. *The above derivation assumes that all questions in the question bank $Q$ are independent, and students with high abilities should perform well on each question.* This formula indicates that for each tested student, answering questions that students with weaker abilities cannot answer correctly enhances ranking consistency.

Considering the asymmetry between collaborative students and tested students, we also need to consider the situation where collaborative students have stronger abilities than tested students:

$$
P(\theta^T < \theta_c^T | \theta^* < \theta_c^*, S) \Rightarrow \sum_{q_i \in S} R(q_i | \theta^* < \theta_c^*),
\tag{6}
$$

where $R(q_i | \theta^* < \theta_c^*) = a_i P(y_i = 0 | \theta^*) P(y_i^c = 1 | \theta^* < \theta_c^*)$. Similar to equation (5), our objective is to shield students from being assessed on questions that students with higher abilities may struggle to answer accurately. By utilizing the constraints from formulas (5) and (6), we can select specific questions for the tested students based on their collaborative students:

$$
q_t = \arg \max_{q \in Q \setminus S_{t-1}} P(\theta^* < \theta_c^*) R(q | \theta^* < \theta_c^*) + P(\theta^* > \theta_c^*) R(q | \theta^* > \theta_c^*).
\tag{7}
$$

Here $S_{t-1}$ represents the subset of questions selected up to step $t$, with $S_t = S_{t-1} \cup \{q_t\}$ where $q_t$ is the question selected at step $t$. This formula aims to find questions that collaborative students with higher abilities are likely to answer correctly, while tested students may struggle with. Meanwhile, it also identifies questions that collaborative students with lower abilities are unlikely to answer correctly, while tested students may respond correctly. The selection method enhances the performance of the originally strong students while diminishing that of weaker ones, aiding tested students in determining their ranking among collaborative students.

After testing, the tested students received their performance on $S$, as well as their ranking relationship with each collaborative student. In the study, we used the mean ranking relationship among collaborative students as the test results for the tested students:

$$
\hat{\theta}^T = \mathbb{E}\left[ I\left(\theta^T > \theta_c^T\right) \right] = \mathbb{E}\left[ I\left( \sum_{i \in S} a_i(y_i - y_i^c) > 0 \right) \right],
\tag{8}
$$

where $I(\cdot)$ is the indicator function. Due to the uncertainty of the tested students' abilities and the incomplete responses from collaborative students during the testing process, we further approximate and elucidate the optimization problem in appendix section C.

---

**Algorithm 1:** The CCAT framework

---

**Require:** $Q$-question bank, $IRT$-estimation method.
**Initialize:** Random initialize tested student's ability $\theta^0$, initialize the question subset $S_t \leftarrow \emptyset$, the tested student's record $Y \leftarrow \emptyset$ and collaborative students' records $\mathbf{Y^c} \leftarrow \emptyset$.

1 **for** $t = 1$ **to** $T$ **do**
2     Select question:
    $q_t \leftarrow \arg\max_{q \in Q \setminus S_{t-1}} P(\theta^* < \theta_c^*) R(q|\theta^* < \theta_c^*) + P(\theta^* > \theta_c^*) R(q|\theta^* > \theta_c^*)$,
    $S_t \leftarrow S_{t-1} \cup \{q_t\}$.
3     Get tested student's and collaborative students' answer:
    $Y \leftarrow Y \cup \{y_t\}, \mathbf{Y^c} \leftarrow \mathbf{Y^c} \cup \{\{y_{1t}^c, \cdots, y_{Mt}^c\}\}$.
4     Update students' estimate ability $\theta$: $\theta^t = \arg\min_{\theta \in \Theta} -\log p_\theta(q_i, y_i)$.
5 Calculate tested student's rank in collaborative students: $\hat{\theta}^T \leftarrow \frac{1}{M} \sum_{i=1}^M \sigma(\sum_{t=1}^T a_i(y_{it}^c - y_t))$.
  **Output:** The student's final estimate ranking ability $\hat{\theta}^T$.

---

### 4.2 Theoretical Analyses of CCAT

Through the above derivation and approximation, we provide the selection algorithm and estimation method for CCAT, which can ensure high degree of consistency in ranking between collaborative and tested students. This ranking is then used to provide the test results for the tested students, denoted as $\hat{\theta}^T$. Regarding the test result $\hat{\theta}^T$ in ability estimation, we have the following conclusion:

**Theorem 1.** *Given two students $A$ and $B$, their relationship with collaborative students are $r_1, r_2, \cdots, r_M; \tilde{r}_1, \tilde{r}_2, \cdots, \tilde{r}_M, r_i \in \{0, 1\}$ indicating whether student $A$ outperforms collaborative student $i$ in a given test. Assuming the probability that student $A$ outperforms the collaborative students $i$ is $P(r_i = 1) = \zeta_1$ and student $B$ outperforms the collaborative students $i$ is $P(\tilde{r}_i = 1) = \zeta_2$. Then the following conclusion can be drawn:*

*(1) If $M > \frac{\ln \frac{1}{\delta}}{2(\zeta_1 - \zeta_2)^2}$ collaborative students are provided, the prediction of ranking consistency will be at least $1 - \delta$.*

*(2) When the number of test questions $T$ is small, the assessment of the ranking relationship between the tested students and collaborative students may yield inaccurate results. Assuming an error probability of $\rho \in (0, 0.5)$, we can still derive that if $M > \frac{\ln \frac{1}{\delta}}{2(1-2\rho)^2(\zeta_1 - \zeta_2)^2}$ collaborative students are provided, the prediction of ranking consistency will be at least $1 - \delta$.*

Drawing from Theorem 1, we can deduce that having a sufficient number of collaborative students ensures a consistent ranking of abilities among all tested students, even in the presence of rank errors between the tested and collaborative students. Meanwhile, Our question selection algorithm actually reduces the ranking error $\rho$ by maximizing the ranking consistency between collaborative and tested students, thereby theoretically increasing the ranking consistency.

Algorithm 1 outlines the pseudo-code for the CCAT framework. During the question selection phase, the complexity of our proposed question selection algorithm is $O(|Q|TMN)$, as it involves selecting the most appropriate question from the question bank $Q$ with a complexity of $O(|Q|M)$ for each tested student. Here, $T$ denotes the total number of questions in the test, $M$ is the number of collaborative students, and $N$ is the number of students being tested. It can be observed that the time complexity of CCAT is comparable to the inference speed of data-driven CAT methods. However, CCAT circumvents the time-consuming training process by storing collaborative students. Although this does increase spatial complexity, it significantly reduces the time required for training and eliminates the need for repeated training of models due to system changes.

## 5 Experiments

In this section, to demonstrate the effectiveness of CCAT on ranking consistency, we compare the performance of CCAT on the ranking consistency metric with other baseline methods on two real-world datasets. In addition, we conduct a case study to compare IRT and collaborative ability estimation and gain deeper insights on how collaborative ability estimation leads to ranking consistency.

### 5.1 Experimental Setup

**Evaluation Method.** The goal is to ensure consistency in the ranking of the test results of tested students on the subsets $S$ and their abilities on all questions in question bank $Q$. In this study, we use the Kendall coefficient [46] between the abilities of tested students on the subsets $S$ and on question bank $Q$, which we call **intra-class ranking consistency**:

$$K = \frac{2}{N(N-1)} \sum_{1 \leq i < j \leq N} U_{ij},$$

$$U_{ij} = \begin{cases} 1 & (\theta_i^* - \theta_j^*)(\theta_i^T - \theta_j^T) \geq 0 \\ 0 & (\theta_i^* - \theta_j^*)(\theta_i^T - \theta_j^T) < 0 \end{cases}. \tag{9}$$

For any two students, if the ranking of their test results aligns with their true abilities, the metric record is 1. Conversely, if the ranking of test results diverges from their true abilities, the metric record is 0.

Similarly, we can also examine the ranking consistency between the tested students and collaborative students, which we refer to as **inter-class ranking consistency**:

$$K = \frac{1}{MN} \sum_{i=1}^{N} \sum_{j=1}^{M} U_{ij}, \tag{10}$$

where $M$ and $N$ are the number of collaborative students and tested students. In addition, we also discuss **AUC**, **ACC** indicators in the main text, and **NDCG** indicator is used as a reference indicator in appendix section D.2 [47, 48, 49].

**Dataset.** We individually conduct experiments on two educational benchmark datasets, NIPS-EDU and JUNYI. NIPS-EDU [50] is a dataset compiled from student question interactions collected from Eedi and used in the NeurIPS 2020 Educational Challenge. JUNYI [51] is sourced from junyiacademy.org, providing millions of response data from students enrolled in a course between 2018 and 2019. The rationale for selecting these two datasets is their extensive student population and the high volume of questions answered by each student, thus facilitating the construction of the collaborative student set. We filter out students who answer less than 50 times and questions that are answered less than 50 times in the following experiment and then divide the dataset into a training dataset (Collaborative Students) and a testing dataset (Tested Students) in a 4:1 ratio. The code can be found in the github: `https://github.com/bigdata-ustc/CCAT`.

**Compared Approaches.** This article primarily focuses on the discussion of ranking consistency in testing, and therefore, we employ IRT, which can provide practical significance results $\theta$. As we know, Monte Carlo sampling (MCMC) and gradient descent (GD) methods can estimate the IRT parameter $a_i, b_i$. In this experiment, we respectively employ the IRT model, estimated by both the GD and MCMC methods, and conduct question selection and student estimation. In terms of the question selection algorithm, we select the following SOTA algorithms as the baseline: **Random** Randomly select a question for students each step, which is a benchmark to evaluate the improvement of other selection algorithms. **FSI** [32] and **KLI** [33] select the question with the highest Fisher/KL information, which measures how much information of students' abilities $\theta$ can be obtained by answering a question. **MAAT** [34] utilizes active learning methods to measure the information each question brings to testing. **BECAT** [40] regards CAT question selection as a coreset selection problem and provides an approximate solution strategy. **BOBCAT** [35] proposed a Bilevel Optimization-based framework to synchronously optimize the question selection algorithm and estimation model. **NCAT** [36] respectively utilizes the ideas of reinforcement learning, and uses data-driven methods to directly optimize the accuracy of CAT test results.

### 5.2 Results and Discussion

To prove the superiority of CCAT framework, we respectively compare various CAT question selection algorithms on IRT estimated by GD and MCMC methods. The following conclusions are obtained:

**Intra-class Ranking Consistency Performance.** Table 1 indicates that Method X consistently enhances ranking consistency at every step after employing collaborative ability estimation (X-C).

Table 1: The Performance of Different Question Selection Algorithms on Intra-class Ranking Consistency. Algorithm **X-C** means use algorithm **X** for question selection but use collaborative ability estimation proposed in CCAT as the testing result instead of the abilities estimated by IRT. CCAT (w/o C) means using the question selection algorithm but estimating the ability by IRT. The bold font represents a significant improvement in statistics compared to the baseline.

(a) Intra-class Ranking Consistency Performance on IRT estimated by GD

| Method Type | Dataset | NIPS-EDU | | | | JUNYI | | | |
|---|---|---|---|---|---|---|---|---|---|
| | Step | 5 | 10 | 15 | 20 | 5 | 10 | 15 | 20 |
| Baseline | BOBCAT | 0.5770 | 0.6362 | 0.6572 | 0.6666 | 0.7104 | 0.7647 | 0.7882 | 0.8044 |
| | Random | 0.7041 | 0.7434 | 0.7680 | 0.7856 | 0.6875 | 0.7350 | 0.7671 | 0.7914 |
| | FSI | 0.7236 | 0.7889 | 0.8192 | 0.8321 | 0.7639 | 0.8284 | 0.8586 | 0.8740 |
| | KLI | 0.7328 | 0.7868 | 0.8142 | 0.8316 | 0.7748 | 0.8340 | 0.8623 | 0.8817 |
| | MAAT | 0.6725 | 0.7095 | 0.7359 | 0.7535 | 0.6908 | 0.7465 | 0.7817 | 0.8113 |
| | NCAT | 0.7611 | 0.8020 | 0.8266 | 0.8359 | 0.5198 | 0.6341 | 0.6803 | 0.7056 |
| | BECAT | 0.7087 | 0.7542 | 0.7802 | 0.7957 | 0.7248 | 0.7712 | 0.7920 | 0.8030 |
| Ours | Random-C | 0.6988 | 0.7444 | 0.7715 | 0.7909 | 0.6862 | 0.7383 | 0.7734 | 0.7979 |
| | FSI-C | 0.7340 | 0.8031 | 0.8339 | **0.8546** | 0.7736 | 0.8313 | 0.8623 | 0.8768 |
| | KLI-C | 0.7399 | 0.7982 | 0.8304 | 0.8509 | 0.7813 | 0.8367 | 0.8671 | 0.8847 |
| | MAAT-C | 0.6689 | 0.7175 | 0.7475 | 0.7603 | 0.7040 | 0.7822 | 0.8222 | 0.8464 |
| | NCAT-C | **0.7691** | 0.8072 | 0.8317 | 0.8412 | 0.5049 | 0.6619 | 0.7194 | 0.7663 |
| | BECAT-C | 0.7292 | 0.7959 | 0.8279 | 0.8438 | 0.7603 | 0.8295 | 0.8603 | 0.8769 |
| | CCAT (w/o C) | 0.7320 | 0.7870 | 0.8177 | 0.8279 | 0.8026 | 0.8560 | 0.8819 | 0.8978 |
| | CCAT | 0.7533 | **0.8081** | **0.8364** | 0.8543 | **0.8092** | **0.8647** | **0.8911** | **0.9066** |

(b) Intra-class Ranking Consistency Performance on IRT estimated by MCMC

| Method Type | Dataset | NIPS-EDU | | | | JUNYI | | | |
|---|---|---|---|---|---|---|---|---|---|
| | Step | 5 | 10 | 15 | 20 | 5 | 10 | 15 | 20 |
| Baseline | Random | 0.7411 | 0.8061 | 0.8348 | 0.8540 | 0.6527 | 0.7759 | 0.8292 | 0.8600 |
| | FSI | 0.7912 | 0.8570 | 0.8846 | 0.8975 | 0.8212 | 0.8820 | 0.9092 | 0.9257 |
| | KLI | 0.7821 | 0.8532 | 0.8804 | 0.8965 | 0.8124 | 0.8795 | 0.9082 | 0.9244 |
| | MAAT | 0.6762 | 0.8083 | 0.8588 | 0.8843 | 0.7404 | 0.8506 | 0.8925 | 0.9161 |
| | NCAT | 0.7766 | 0.8451 | 0.8710 | 0.8831 | 0.7430 | 0.8203 | 0.8526 | 0.8737 |
| | BECAT | 0.7685 | 0.8441 | 0.8766 | 0.8958 | 0.7857 | 0.8699 | 0.9031 | 0.9225 |
| Ours | Random-C | 0.7531 | 0.8084 | 0.8363 | 0.8547 | 0.7511 | 0.8074 | 0.8429 | 0.8667 |
| | FSI-C | 0.7933 | 0.8573 | 0.8848 | **0.8977** | 0.8226 | 0.8820 | 0.9090 | 0.9251 |
| | KLI-C | 0.7839 | 0.8530 | 0.8805 | 0.8966 | 0.8146 | 0.8795 | 0.9079 | 0.9237 |
| | MAAT-C | 0.6909 | 0.8090 | 0.8595 | 0.8848 | 0.7441 | 0.8512 | 0.8926 | 0.9157 |
| | NCAT-C | 0.7923 | 0.8501 | 0.8725 | 0.8840 | 0.7829 | 0.8359 | 0.8615 | 0.8784 |
| | BECAT-C | 0.7680 | 0.8449 | 0.8771 | 0.8961 | 0.7932 | 0.8706 | 0.9027 | 0.9217 |
| | CCAT (w/o C) | 0.7982 | 0.8561 | 0.8832 | 0.8955 | 0.8190 | 0.8823 | 0.9098 | **0.9277** |
| | CCAT | **0.8149** | **0.8635** | **0.8851** | 0.8969 | **0.8448** | **0.8875** | **0.9100** | 0.9273 |

This finding aligns with Theorem 1, which subst antiates the effectiveness of collaborative ability estimation in CCAT. Furthermore, when comparing Method X-C, whether employing MCMC or GD methods for estimating IRT model parameters, our CCAT algorithm demonstrates superior performance in ranking consistency across two public datasets. Particularly, CCAT shows more significant improvement when fewer questions are tested, outperforming other methods. As the number of test steps increases, the FSI-C method improves ranking consistency more rapidly, ultimately achieving a high level of consistency. This is attributed to the FSI method's ability to select questions with higher discrimination and uncertain responses, enabling the FSI-C method to promptly adjust students with inconsistent ranking. However, due to the FSI method's sensitivity to current abilities, it performs inadequately when fewer questions are tested. These results confirm that the CCAT framework is generally effective in ranking for CAT, whether in terms of test duration or estimation model.

**Inter-class Ranking Consistency Performance.** After each baseline selection algorithm is completed, we replace the original results obtained by directly using IRT for parameter estimation with the ranking results obtained from collaborative ability estimation. From Tables 1 and 2, it can be seen that there is a positive correlation between the ranking consistency of the tested students among the collaborative students (Table 2) and the ranking consistency among the tested students (Table

Table 2: Inter-class Ranking Consistency Performance on IRT estimated by MCMC, which measures the accuracy of the collaborative ability estimation.

| Dataset | NIPS-EDU | | | | JUNYI | | | |
|---|---|---|---|---|---|---|---|---|
| Step | 5 | 10 | 15 | 20 | 5 | 10 | 15 | 20 |
| Random | 0.7798 | 0.8325 | 0.8590 | 0.8760 | 0.7651 | 0.8298 | 0.8648 | 0.8865 |
| FSI | 0.8258 | 0.8785 | 0.9013 | **0.9126** | 0.8575 | 0.9050 | 0.9249 | 0.9363 |
| KLI | 0.8195 | 0.8758 | 0.8985 | 0.9119 | 0.8502 | 0.9028 | 0.9240 | 0.9353 |
| MAAT | 0.7242 | 0.8373 | 0.8807 | 0.9023 | 0.7830 | 0.8767 | 0.9069 | 0.9249 |
| NCAT | 0.8286 | 0.8697 | 0.8892 | 0.8994 | 0.8090 | 0.8604 | 0.8830 | 0.8972 |
| BECAT | 0.8045 | 0.8676 | 0.8948 | 0.9104 | 0.8287 | 0.8961 | 0.9204 | 0.9341 |
| CCAT | **0.8476** | **0.8839** | 0.9013 | 0.9116 | **0.8736** | **0.9082** | **0.9255** | **0.9373** |

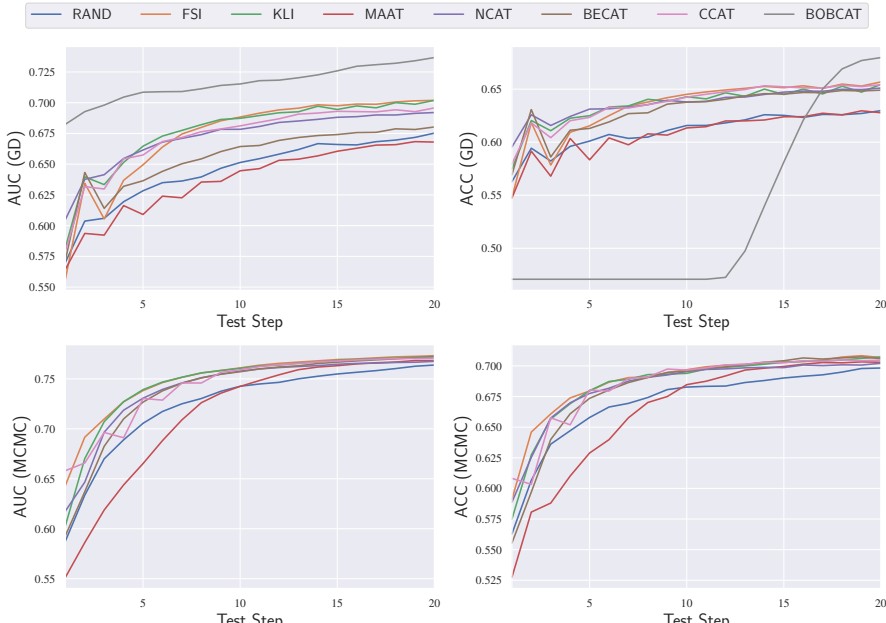

Figure 3: The performance on ACC and AUC of different question selection algorithms on the dataset NIPS-EDU for the IRT model estimated by MCMC and GD methods.

1) when using the collaborative ability estimation method, especially, CCAT method is in a leading position in both two tables, and FSI method is only second to CCAT. This also explains why we optimize the ranking consistency among the collaborative students in the above section.

**ACC&AUC.** Figure 3 displays the metrics (ACC, AUC) obtained through various question selection algorithms on IRT, as estimated by different methods. It is evident that CCAT, when compared to other CAT question selection algorithms, does not significantly differ in terms of ACC and AUC indicators. This suggests that CCAT maintains the accuracy of CAT test results while enhancing ranking consistency. Furthermore, IRT estimated by MCMC significantly outperforms that estimated by GD and BOBCAT. This also explains why the same question selection algorithm in Table 1 performs better on the IRT model obtained through MCMC. Additionally, question selection algorithms proposed on GD, particularly those such as NCAT that utilize data-driven methods, are not efficient for IRT estimated by MCMC. This implies that these methods may not be effective, but can compensate for the drawbacks of using GD to estimate IRT. However, methods like BOBCAT, which concurrently train the IRT model alongside a question selection algorithm, may introduce bias into the IRT model. As depicted in Figure 3, while it outperforms all gradient descent methods in specific optimization objectives (ACC@20), it may impact accuracy at other times and compromise the stability of the IRT model in ability estimation. This can result in suboptimal performance in ranking problems. Given the analysis above and the stability of the MCMC method, we assert that it

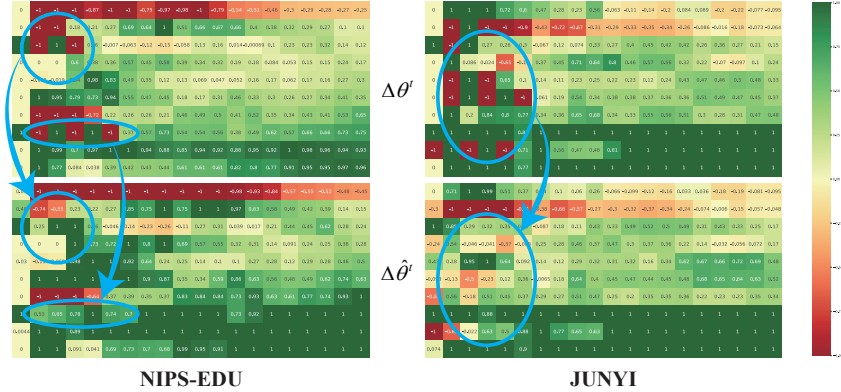

$\Delta\theta^t$

$\Delta\hat{\theta}^t$

**NIPS-EDU**              **JUNYI**

Figure 4: Visualization of differences in abilities estimation by IRT method and CCAT method

is more appropriate for IRT parameter estimation than the GD method, particularly when considering the ranking consistency of CAT.

**Case Study.** To demonstrate the superiority of CCAT and its mechanism, we select 10 student pairs from each dataset and conduct two visualization experiments as shown in Figures 4. This figure compares the ability gap between student pairs as estimated by the IRT and CCAT methods. Specifically, for each student pair, we subtract the estimated ability of the student with higher true ability from that of the student with lower true ability at each moment. A larger gap indicates better discrimination by the estimation method. When the value is less than 0 (red), it signals a ranking inconsistency at that point in time. Our findings show that, although the selection algorithm remains unchanged, CCAT produces greater discrimination and more accurate rankings, particularly when fewer testing steps are involved.

We also analyze the estimation results for collaborative students on these 20 student pairs, revealing that the collaborative ability estimation method essentially functions as a voting process by collaborative students for the tested students. Additionally, we visualize how each collaborative student's judgment of the two students becomes progressively clearer as the number of test questions increases. For further details, please refer to Appendix D.2.

# 6 Conclusion

This article explored the objectives of Computerized Adaptive Testing (CAT) from the perspective of students, reframing CAT challenges as ranking tasks and proposing specific objectives for these tasks. To address the challenge of students working independently, which limits influence on rankings during the testing process, we introduced a Collaborative Computerized Adaptive Testing (CCAT) framework. This approach leverages collaborative student interactions to assist in question selection and estimation during testing. Experiments on two real-world datasets demonstrated that CCAT improves ranking consistency. Despite these promising results, our method has inherent limitations, particularly with longer testing sequences. In future work, we aim to refine our model to address these limitations and enhance the robustness and effectiveness of the CCAT framework across diverse testing scenarios.

**Acknowledgments and Disclosure of Funding**

This research was supported by grants from the National Key Research and Development Program of China (Grant No. 2021YFF0901003), the Key Technologies R & D Program of Anhui Province (No. 202423k09020039), the University Synergy Innovation Program of Anhui Province (GXXT-2022-042) and the Fundamental Research Funds for the Central Universities.

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

# A    Proofs of Lemma 1

**Lemma 1.** *Given two students, whose responses on $S(|S| = T)$ are $y_1, y_2, \cdots, y_T$ and $\tilde{y}_1, \tilde{y}_2, \cdots, \tilde{y}_T$, their testing abilities on $S$ are $\theta^T, \tilde{\theta}^T$, which are estimated by IRT with parameters $a_i, b_i$. We can prove that if $(\theta^T - \tilde{\theta}^T) > 0$, then $(\sum_{i=1}^{T} a_i(y_i - \tilde{y}_i)) > 0$, vice versa.*

*Proof.* Since the abilities of $\theta^T$ and $\tilde{\theta}^T$ are the maximum likelihood estimation of IRT in Formula 2, they meet the following conditions:

$$\frac{\partial \ln L}{\partial \theta} = \sum_{i=1}^{T} a_i(P_i(\theta^T) - y_i) = \sum_{i=1}^{T} a_i(P_i(\tilde{\theta}^T) - \tilde{y}_i) = 0.$$

According to the Lagrange mean value theorem [52], the following derivation can be derived:

$$\sum_{i=1}^{T} a_i(y_i - \tilde{y}_i) = \sum_{i=1}^{T} a_i(P_i(\theta^T) - P_i(\tilde{\theta}^T)) = \sum_{i=1}^{T} a_i P_i'(\zeta_i)(\theta^T - \tilde{\theta}^T).$$

Since $P_i'(\zeta_i) = a_i P_i(\zeta_i)(1 - P(\zeta_i))$ and $0 < P_i(\zeta_i) < 1$, it implies that:

$$\sum_{i=1}^{T} a_i(y_i - \tilde{y}_i) = (\sum_{i=1}^{T} a_i^2 P(\zeta)(1 - P(\zeta)))(\theta^T - \tilde{\theta}^T).$$

Due to $\sum_{i=1}^{T} a_i(y_i - \tilde{y}_i)$ and $\theta^T - \tilde{\theta}^T$ shared positivity or negativity:

$$\sum_{i=1}^{T} a_i(y_i - \tilde{y}_i) > 0 \Leftrightarrow \theta^T - \tilde{\theta}^T > 0 \qquad \square$$

# B    Proofs of Theorem 1

**Theorem 1.** *Given two students A and B, their relationship with collaborative students are $r_1, r_2, \cdots, r_M; \tilde{r}_1, \tilde{r}_2, \cdots, \tilde{r}_M, r_i \in \{0, 1\}$ indicating whether student A outperforms collaborative student $i$ in a given test. Assuming the probability that student A outperforms the collaborative students $i$ is $P(r_i = 1) = \zeta_1$ and student B outperforms the collaborative students $i$ is $P(\tilde{r}_i = 1) = \zeta_2$. Then the following conclusion can be drawn:*

*(1) If $M > \frac{\ln \frac{1}{\delta}}{2(\zeta_1 - \zeta_2)^2}$ collaborative students are provided, the prediction of ranking consistency will be at least $1 - \delta$.*

*(2) When the number of test questions $T$ is small, the assessment of the ranking relationship between the tested students and collaborative students may yield inaccurate results. Assuming an error probability of $\rho \in (0, 0.5)$, we can still derive that if $M > \frac{\ln \frac{1}{\delta}}{2(1-2\rho)^2(\zeta_1 - \zeta_2)^2}$ collaborative students are provided, the prediction of ranking consistency will be at least $1 - \delta$.*

*Proof.* Assuming that the ranking abilities of two students are $\hat{\theta}_1^T, \hat{\theta}_2^T$. Without loss of generality, we suppose $\zeta_1 > \zeta_2$. We define the random variable $X_i$ as the relationship between two students' ranking, where $X_i = r_i - \tilde{r}_i$.

Suppose, we have $M$ collaborative students, and we define $\overline{X} = \frac{1}{M} \sum_{i=1}^{M} X_i$. Obviously,

$$\mathbb{E}[\hat{\theta}_1^T - \hat{\theta}_2^T] = \mathbb{E}\overline{X} = \frac{1}{M} \sum_{i=1}^{M} \mathbb{E}X_i = (1-\rho)\zeta_1 + \rho(1-\zeta_1) - (1-\rho)\zeta_2 - \rho(1-\zeta_2) = (1-2\rho)(\zeta_1 - \zeta_2).$$

According to the Hoeffding's inequality [53, 54], we have:

$$P(\hat{\theta}_1^T < \hat{\theta}_2^T) = P(\hat{\theta}_1^T - \hat{\theta}_2^T - \mathbb{E}[\hat{\theta}_1^T - \hat{\theta}_2^T] < -(1-2\rho)(\zeta_1 - \zeta_2))) < exp(-2M[(1-2\rho)(\zeta_1 - \zeta_2)]^2).$$

Setting $\delta = exp(-2M(1 - 2\rho)^2(\zeta_1 - \zeta_2)^2)$, we have when $M$ is larger than $\frac{\ln \frac{1}{\delta}}{2(1-2\rho)^2(\zeta_1 - \zeta_2)^2}$, $P(\hat{\theta}_1^T < \hat{\theta}_2^T) < \delta$, which means the prediction error is small than $\delta$. $\qquad \square$

## C   Implementation Details of CCAT

Due to the incomplete information in the test, we also made the following approximations:

**(1) Approximate Collaborative Students.**   Since there is no collaborative student in the real world who has completely completed all the questions as we assumed, we use $P_i(\theta_c^*)$ to supplement the answer for question $i$, which means:

$$y'^c_i = \begin{cases} 1 & y_i^c = 1 \\ 0 & y_i^c = 0 \\ P_i(\theta_c^*) & y_i^c = None \end{cases}. \tag{11}$$

Based on this, if $y_i^c$ is not provided, $I(y_i^c = 1)$ should be replaced to $P_i(\theta_c^*)$ and $I(y_i^c = 0)$ should be replaced to $1 - P_i(\theta_c^*)$ in question selection part, and in ability estimation part, $I\left(\sum_{i \in S} a_i(y_i - y_i^c) > 0\right)$ can be approximated as $\sigma\left(\sum_{i \in S} a_i(y_i - y'^c_i)\right)$ and it can be applied regardless of whether there is $P_i(\theta_c^*)$ of supplementation or not.

**(2) Approximate Outperform Probability.**   In our method, we need to select questions by using the information on whether tested students outperform the collaborative students $I(\theta^* > \theta_c^*)$. However, the ground truth $\theta^*$ and $\theta_c^*$ is unknown when testing. So we proposed using $\theta^t$ and $\theta_c^t$ to approximate $\theta^*$ and $\theta_c^*$ for each step $t$. Considering that there is a certain error between time t and the actual state, we use the sigmoid function $\sigma(\theta^t - \theta_c^t)$ to approximate $I(\theta^* > \theta_c^*)$, which means the more tested students are ahead of collaborative students at step $t$, the higher the likelihood that their true abilities surpass those of the collaborative students. Through the above approximation, the question selection algorithm can be rewritten as follows:

$$i_t = \arg \max_{q_i \in Q \backslash S_{t-1}} \tilde{R}(q_i, \theta^* > \theta_c^* | \theta^{t-1}) + \tilde{R}(q_i, \theta^* < \theta_c^* | \theta^{t-1}). \tag{12}$$

where $\tilde{R}(q_i, \theta^* > \theta_c^* | \theta^{t-1}) = a_i P(y_i = 0 | \theta^t) \left[\sum_{y'^c \in C} y'^c_i \sigma\left(\sum_{i \in S} a_i(y_i - y'^c_i)\right)\right]$ and $\tilde{R}(q_i, \theta^* < \theta_c^* | \theta^{t-1}) = a_i P(y_i = 1 | \theta^t) \left[\sum_{y'^c \in C} (1 - y'^c_i)\sigma\left(\sum_{i \in S} a_i(y'^c_i - y_i)\right)\right]$, $P(y_i = 0 | \theta^t), P(y_i = 1 | \theta^t)$ can be calculated by IRT method and $C$ is the set of collaborative students.

## D   Details of Experiments

### D.1   Statistics of the datasets.

Table 3: Statistics of the datasets

| Dataset | **NIPS-EDU** | **JUNYI** |
|---|---|---|
| #Students | 4,914 | 8,852 |
| #Questions | 900 | 702 |
| #Response logs | 1,382,173 | 801,270 |
| #Response logs per student | 281.27 | 90.52 |
| #Response logs per question | 1,535.75 | 1,141.41 |

### D.2   Detailed Evaluation Method

**Statistic for Ranking Consistency.**   For CAT tasks, there are many methods that are sensitive to the initial abilities of students, including Random, FSI, KLI, MAAT, BECAT, and CCAT proposed in this article. However, data-driven methods such as BOBCAT and NCAT are often insensitive to the initial abilities of students. Therefore, this study randomly initialized the initial abilities of students 5 times and counted the mean and standard deviation of the ranking consistency of each question selection algorithm, as shown in Tables 4 and 5. It can be seen that although the current abilities of students are used in the selection process, CCAT is almost not affected by the initialization of student abilities. This indicates that CCAT not only performs well in ranking consistency but also is more stable compared to other strategies.

Table 4: The Detail Performance of different question selection algorithms on **NIPS-EDU**. Algorithm **X-C** means use algorithm **X** for question selection but use collaborative ability estimation proposed in CCAT as the testing result instead of the abilities estimated by IRT. The bold font represents a significant improvement in statistics compared to the baseline.

(a) Intra-class Ranking Consistency Performance on IRT estimated by GD

| Dataset | NIPS-EDU | | | |
|---|---|---|---|---|
| Step | 5 | 10 | 15 | 20 |
| Random | $0.7041 \pm 0.007$ | $0.7434 \pm 0.005$ | $0.7680 \pm 0.007$ | $0.7856 \pm 0.004$ |
| FSI | $0.7236 \pm 0.004$ | $0.7889 \pm 0.003$ | $0.8192 \pm 0.002$ | $0.8321 \pm 0.002$ |
| KLI | $0.7328 \pm 0.004$ | $0.7868 \pm 0.005$ | $0.8142 \pm 0.003$ | $0.8316 \pm 0.002$ |
| MAAT | $0.6725 \pm 0.001$ | $0.7095 \pm 0.002$ | $0.7359 \pm 0.002$ | $0.7535 \pm 0.001$ |
| BECAT | $0.7087 \pm 0.007$ | $0.7542 \pm 0.004$ | $0.7802 \pm 0.005$ | $0.7957 \pm 0.005$ |
| CCAT (w/o C) | $0.7320 \pm 0.002$ | $0.7870 \pm 0.002$ | $0.8177 \pm 0.002$ | $0.8279 \pm 0.002$ |
| Random-C | $0.6988 \pm 0.008$ | $0.7444 \pm 0.004$ | $0.7715 \pm 0.005$ | $0.7909 \pm 0.004$ |
| FSI-C | $0.7340 \pm 0.005$ | $0.8031 \pm 0.003$ | $0.8339 \pm 0.002$ | $\mathbf{0.8546} \pm 0.001$ |
| KLI-C | $0.7399 \pm 0.003$ | $0.7982 \pm 0.003$ | $0.8304 \pm 0.002$ | $0.8509 \pm 0.001$ |
| MAAT-C | $0.6689 \pm 0.002$ | $0.7175 \pm 0.003$ | $0.7475 \pm 0.002$ | $0.7603 \pm 0.002$ |
| BECAT-C | $0.7292 \pm 0.006$ | $0.7959 \pm 0.003$ | $0.8279 \pm 0.002$ | $0.8438 \pm 0.007$ |
| CCAT | $\mathbf{0.7533} \pm 0.000$ | $\mathbf{0.8081} \pm 0.001$ | $\mathbf{0.8364} \pm 0.000$ | $0.8543 \pm 0.000$ |

(b) Intra-class Ranking Consistency Performance on IRT estimated by MCMC

| Dataset | NIPS-EDU | | | |
|---|---|---|---|---|
| Step | 5 | 10 | 15 | 20 |
| Random | $0.7411 \pm 0.005$ | $0.8061 \pm 0.005$ | $0.8348 \pm 0.004$ | $0.8540 \pm 0.005$ |
| FSI | $0.7912 \pm 0.005$ | $0.8570 \pm 0.003$ | $0.8846 \pm 0.001$ | $0.8975 \pm 0.001$ |
| KLI | $0.7821 \pm 0.005$ | $0.8532 \pm 0.003$ | $0.8804 \pm 0.001$ | $0.8965 \pm 0.002$ |
| MAAT | $0.6762 \pm 0.005$ | $0.8083 \pm 0.007$ | $0.8588 \pm 0.004$ | $0.8843 \pm 0.002$ |
| BECAT | $0.7685 \pm 0.005$ | $0.8441 \pm 0.002$ | $0.8766 \pm 0.002$ | $0.8958 \pm 0.002$ |
| CCAT (w/o C) | $0.7982 \pm 0.001$ | $0.8561 \pm 0.001$ | $0.8832 \pm 0.001$ | $0.8955 \pm 0.000$ |
| Random-C | $0.7531 \pm 0.004$ | $0.8084 \pm 0.005$ | $0.8363 \pm 0.004$ | $0.8547 \pm 0.004$ |
| FSI-C | $0.7933 \pm 0.005$ | $0.8573 \pm 0.003$ | $0.8848 \pm 0.001$ | $\mathbf{0.8977} \pm 0.001$ |
| KLI-C | $0.7839 \pm 0.006$ | $0.8530 \pm 0.003$ | $0.8805 \pm 0.001$ | $0.8966 \pm 0.002$ |
| MAAT-C | $0.6909 \pm 0.005$ | $0.8090 \pm 0.003$ | $0.8595 \pm 0.004$ | $0.8848 \pm 0.002$ |
| BECAT-C | $0.7680 \pm 0.004$ | $0.8449 \pm 0.001$ | $0.8771 \pm 0.002$ | $0.8961 \pm 0.001$ |
| CCAT | $\mathbf{0.8149} \pm 0.002$ | $\mathbf{0.8635} \pm 0.001$ | $\mathbf{0.8851} \pm 0.001$ | $0.8969 \pm 0.000$ |

(c) Inter-class Ranking Consistency Performance on IRT estimated by MCMC

| Dataset | NIPS-EDU | | | |
|---|---|---|---|---|
| Step | 5 | 10 | 15 | 20 |
| Random | $0.7798 \pm 0.003$ | $0.8325 \pm 0.003$ | $0.8590 \pm 0.002$ | $0.8760 \pm 0.002$ |
| FSI | $0.8258 \pm 0.003$ | $0.8785 \pm 0.002$ | $0.9013 \pm 0.001$ | $\mathbf{0.9126} \pm 0.001$ |
| KLI | $0.8195 \pm 0.003$ | $0.8758 \pm 0.002$ | $0.8985 \pm 0.001$ | $0.9119 \pm 0.001$ |
| MAAT | $0.7242 \pm 0.004$ | $0.8373 \pm 0.002$ | $0.8807 \pm 0.002$ | $0.9023 \pm 0.001$ |
| BECAT | $0.8045 \pm 0.003$ | $0.8676 \pm 0.001$ | $0.8948 \pm 0.001$ | $0.9104 \pm 0.001$ |
| CCAT | $\mathbf{0.8476} \pm 0.001$ | $\mathbf{0.8839} \pm 0.000$ | $\mathbf{0.9013} \pm 0.000$ | $0.9116 \pm 0.000$ |

Table 5: The Detail Performance of different question selection algorithms on **JUNYI**. Algorithm **X-C** means use algorithm **X** for question selection but use collaborative ability estimation proposed in CCAT as the testing result instead of the abilities estimated by IRT. The bold font represents a significant improvement in statistics compared to the baseline.

(a) Intra-class Ranking Consistency Performance on IRT estimated by GD

| Dataset | JUNYI | | | |
|---|---|---|---|---|
| Step | 5 | 10 | 15 | 20 |
| Random | $0.6875 \pm 0.008$ | $0.7350 \pm 0.005$ | $0.7671 \pm 0.003$ | $0.7914 \pm 0.003$ |
| FSI | $0.7639 \pm 0.004$ | $0.8284 \pm 0.003$ | $0.8586 \pm 0.002$ | $0.8740 \pm 0.002$ |
| KLI | $0.7748 \pm 0.002$ | $0.8340 \pm 0.001$ | $0.8623 \pm 0.001$ | $0.8817 \pm 0.001$ |
| MAAT | $0.6908 \pm 0.000$ | $0.7465 \pm 0.000$ | $0.7817 \pm 0.000$ | $0.8113 \pm 0.000$ |
| BECAT | $0.7248 \pm 0.003$ | $0.7712 \pm 0.003$ | $0.7920 \pm 0.003$ | $0.8030 \pm 0.003$ |
| CCAT (w/o C) | $0.8026 \pm 0.001$ | $0.8560 \pm 0.001$ | $0.8819 \pm 0.000$ | $0.8978 \pm 0.000$ |
| Random-C | $0.6862 \pm 0.008$ | $0.7383 \pm 0.007$ | $0.7734 \pm 0.004$ | $0.7979 \pm 0.003$ |
| FSI-C | $0.7736 \pm 0.005$ | $0.8313 \pm 0.003$ | $0.8623 \pm 0.002$ | $0.8768 \pm 0.002$ |
| KLI-C | $0.7813 \pm 0.001$ | $0.8367 \pm 0.002$ | $0.8671 \pm 0.001$ | $0.8847 \pm 0.002$ |
| MAAT-C | $0.7040 \pm 0.000$ | $0.7822 \pm 0.000$ | $0.8222 \pm 0.000$ | $0.8464 \pm 0.000$ |
| BECAT-C | $0.7603 \pm 0.003$ | $0.8295 \pm 0.002$ | $0.8603 \pm 0.002$ | $0.8769 \pm 0.001$ |
| CCAT | $\mathbf{0.8092} \pm 0.001$ | $\mathbf{0.8647} \pm 0.000$ | $\mathbf{0.8911} \pm 0.000$ | $\mathbf{0.9066} \pm 0.000$ |

(b) Intra-class Ranking Consistency Performance on IRT estimated by MCMC

| Dataset | JUNYI | | | |
|---|---|---|---|---|
| Step | 5 | 10 | 15 | 20 |
| Random | $0.6527 \pm 0.002$ | $0.7759 \pm 0.005$ | $0.8292 \pm 0.002$ | $0.8600 \pm 0.002$ |
| FSI | $0.8212 \pm 0.003$ | $0.8820 \pm 0.001$ | $0.9092 \pm 0.001$ | $0.9257 \pm 0.000$ |
| KLI | $0.8124 \pm 0.003$ | $0.8795 \pm 0.002$ | $0.9082 \pm 0.001$ | $0.9244 \pm 0.001$ |
| MAAT | $0.7404 \pm 0.008$ | $0.8506 \pm 0.001$ | $0.8925 \pm 0.001$ | $0.9161 \pm 0.001$ |
| BECAT | $0.7857 \pm 0.002$ | $0.8699 \pm 0.001$ | $0.9031 \pm 0.001$ | $0.9225 \pm 0.000$ |
| CCAT (w/o C) | $0.8190 \pm 0.001$ | $0.8823 \pm 0.001$ | $0.9098 \pm 0.000$ | $\mathbf{0.9277} \pm 0.000$ |
| Random-C | $0.7511 \pm 0.005$ | $0.8074 \pm 0.005$ | $0.8429 \pm 0.002$ | $0.8667 \pm 0.002$ |
| FSI-C | $0.8226 \pm 0.002$ | $0.8820 \pm 0.001$ | $0.9090 \pm 0.001$ | $0.9251 \pm 0.001$ |
| KLI-C | $0.8146 \pm 0.002$ | $0.8795 \pm 0.002$ | $0.9079 \pm 0.001$ | $0.9237 \pm 0.001$ |
| MAAT-C | $0.7441 \pm 0.003$ | $0.8512 \pm 0.001$ | $0.8926 \pm 0.001$ | $0.9157 \pm 0.001$ |
| BECAT-C | $0.7932 \pm 0.002$ | $0.8706 \pm 0.001$ | $0.9027 \pm 0.001$ | $0.9217 \pm 0.000$ |
| CCAT | $\mathbf{0.8448} \pm 0.007$ | $\mathbf{0.8875} \pm 0.000$ | $\mathbf{0.9100} \pm 0.000$ | $0.9273 \pm 0.000$ |

(c) Inter-class Ranking Consistency Performance on IRT estimated by MCMC

| Dataset | JUNYI | | | |
|---|---|---|---|---|
| Step | 5 | 10 | 15 | 20 |
| Random | $0.7651 \pm 0.003$ | $0.8298 \pm 0.004$ | $0.8648 \pm 0.001$ | $0.8865 \pm 0.001$ |
| FSI | $0.8575 \pm 0.001$ | $0.9050 \pm 0.000$ | $0.9249 \pm 0.000$ | $0.9363 \pm 0.000$ |
| KLI | $0.8502 \pm 0.002$ | $0.9028 \pm 0.001$ | $0.9240 \pm 0.000$ | $0.9353 \pm 0.000$ |
| MAAT | $0.7830 \pm 0.002$ | $0.8767 \pm 0.001$ | $0.9069 \pm 0.001$ | $0.9249 \pm 0.001$ |
| BECAT | $0.8287 \pm 0.001$ | $0.8961 \pm 0.001$ | $0.9204 \pm 0.001$ | $0.9341 \pm 0.000$ |
| CCAT | $\mathbf{0.8736} \pm 0.001$ | $\mathbf{0.9082} \pm 0.000$ | $\mathbf{0.9255} \pm 0.000$ | $\mathbf{0.9373} \pm 0.000$ |

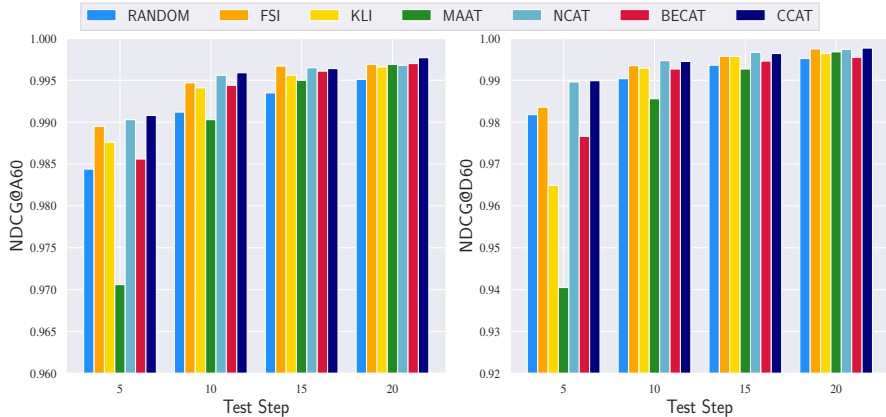

Figure 5: The performance on NDCG of different question selection algorithms on the dataset NIPS-EDU for the IRT model estimated by MCMC method.

**NDCG.** NDCG[55, 56, 57], as an important metric for ranking problems in recommendation systems, is also used as a reference metric for CAT ranking problems. Specifically, at each moment of the test, CAT provides students with an ability estimation, while selection exams can be seen as a recall of students. Specifically, we assume that 60% of students will be admitted or eliminated, which means recalling the top 60% of students (NDCG@A60%) and recalling the bottom 60% of students (NDCG@D60%). From Figure 5, it can be seen that CCAT, as a CAT method proposed for ranking problems, also performs outstandingly in recall tasks, indicating that the CCAT method can provide a more fair selection for selective exams.

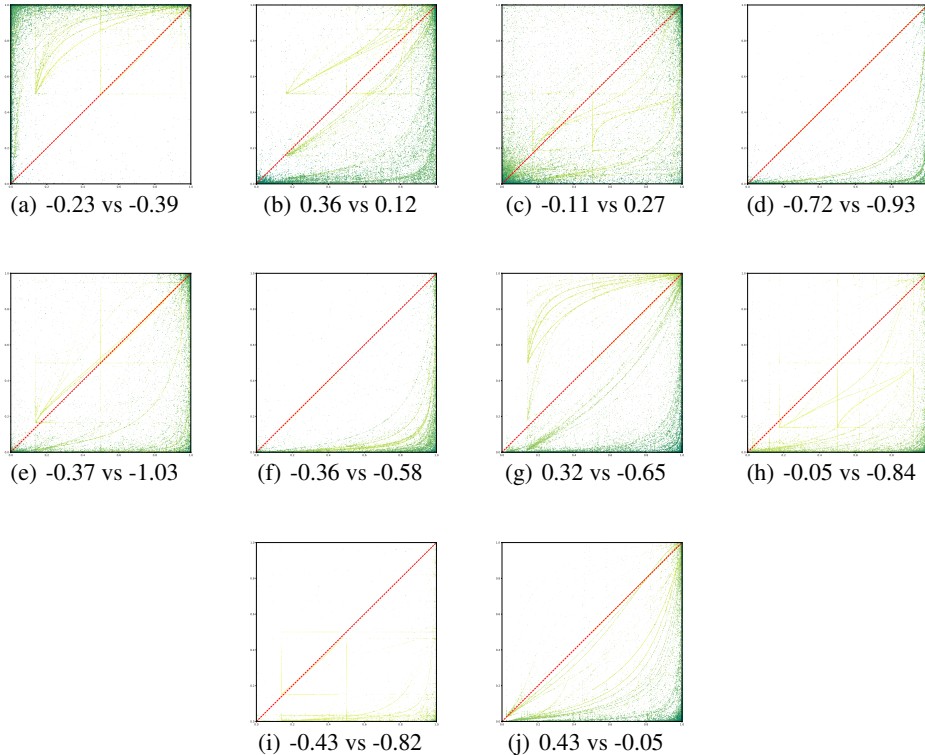

Figure 6: Rating Chart for different students pair estimated by collaborative students in NIPS-EDU.

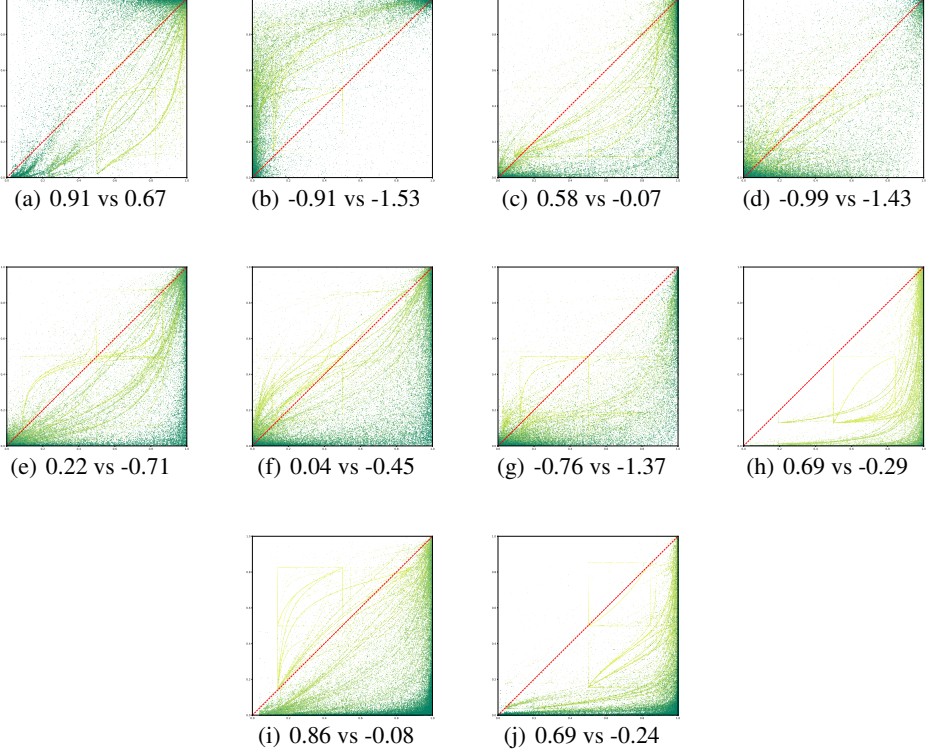

Figure 7: Rating Chart for different students pair estimated by collaborative students in JUNYI.

**Case Study Supplement.** Figures 6 and 7 illustrate the responses of collaborative students within each pair. Each point's coordinates denote the comparative performance of the student pairs relative to individual collaborative students. The intensity of the point's color corresponds to the response time, with darker hues indicating later responses.

Based on Figure 6 and 7, it can be seen that CCAT determines the ranking of students at each moment by comparing the number of collaborative students in the upper and lower triangles. The light-colored points in the figure are mainly distributed in the center, while the dark ones are distributed around, indicating that as the number of test questions increases, each collaborative student's judgment of the two students gradually changes from vague to clear. It can be found that the collaborative ability estimation method is essentially collaborative student voting for tested students, and the collaborative student union in the upper left or lower right corner of the figure will ultimately distinguish the two students.

