# OpenReview forum: "Computerized Adaptive Testing via Collaborative Ranking"
_NeurIPS.cc/2024/Conference — NeurIPS 2024 poster_

### Official Review · Reviewer_gvzk · 2024-07-10

**Soundness:** 4
**Presentation:** 3
**Contribution:** 3
**Rating:** 8
**Confidence:** 4

**Summary:**

In this paper, the authors first discovered the inconsistency issue in existing Computerized Adaptive Testing (CAT) solutions for estimating the latent abilities of students, that is the higher accuracy of ability estimation (with a lower MSE) does not necessarily guarantee the ranking consistency of students’ abilities. Based on this discovery, the authors then proposed a novel Computerized Adaptive Testing framework CCAT inspired by collaborative ranking. Specifically, CCAT uses collaborative students as anchors to assist in test-question selection and estimation in testing. More importantly, the authors provide a theoretical analysis of the upper bound of ranking consistency error for collaborative students, which verifies that with an adequate number of collaborative students, the ranking consistency error can be reduced to an acceptable level. Through experiments on two real-world datasets, the authors demonstrated that CCAT can achieve the best ranking consistency.

**Strengths:**

1. Computerized Adaptive Testing is a cross-cutting research direction between artificial intelligence and the testing area, with broad applications in the real world. In this paper, the authors first discovered the inconsistency issue in existing Computerized Adaptive Testing (CAT) solutions for estimating the latent abilities of students, and therefore, the research motivation is both original and significant.

2. The proposed Collaborative Computerized Adaptive Testing framework CCAT exploits the idea of collaborative students to address the incomparable test-answering behavior problem of different students. This idea is quite novel and differs from traditional CAT solutions and collaborative ranking solutions.

3. Both the theoretical analysis and the experimental validation seem to be solid and convincing.

**Weaknesses:**

1. It is not easy to illustrate the main ideas of CAT as it contains both the Ability Estimation Part and the Question Selection Part. Therefore, the readability of this paper can be further improved especially for the readers without any background of CAT.

2. The main idea of the CCAT algorithm can be explained in more detail in the main text. For instance, how can we get the collaborative students?

3. More references about collaborative ranking are recommended to be included and discussed.

**Questions:**

1. How can we get the collaborative students? Are these collaborative students the same as all the testing students?

2. From Equation (5), the response of the students belongs to 0 or 1, which means the student's responses studied in this paper are either right or wrong, what if the rating of the student responses has more choices (like the values in the range of [0,1])? Does the CCAT solution still work in this scenario?

3. How to determine the parameter T (testing round) in CCAT?

**Limitations:**

The authors have adequately addressed the limitations

---

> ### Author Rebuttal · Authors · 2024-08-05
>
> Thank you for your feedback on our manuscript. We sincerely appreciate your time and effort in evaluating our work, and we appreciate you for the opportunity to explain and articulate our work.
>
> > **Q1:** It is not easy to illustrate the main ideas of CAT as it contains both the Ability Estimation Part and the Question Selection Part. Therefore, the readability of this paper can be further improved especially for the readers without any background of CAT.
> >
> > **Q2:** The main idea of the CCAT algorithm can be explained in more detail in the main text.
>
> We appreciate your observation regarding the difficulty in illustrating the main ideas of CAT, particularly as it involves both the Ability Estimation Part and the Question Selection Part. To address this, we will optimize the content of our article to enhance its readability, especially for readers who do not have a background in CAT. We will ensure that our explanations are more accessible and clear, incorporating additional context and examples where necessary.
>
> > **Q3:** How can we get the collaborative students?
> >
> > **Q4:** Are these collaborative students the same as all the testing students?
>
> In the experimental phase, collaborative students can be constructed by splitting the dataset and supplementing it with the predicted values from the IRT, as detailed in Appendix Section C. We believe that there is no essential difference between these collaborative students and testing students, only that collaborative students have answer records while testing students do not. In real exam scenarios, generally speaking, in real CAT systems such as GRE, a group of students will take tests in advance to generate test records[1]. Moreover, we can also use students who have previously taken exams as collaborative students.
>
> > **Q5:** More references about collaborative ranking are recommended to be included and discussed.
>
> We appreciate the importance of providing a comprehensive context for our work and will include additional references related to collaborative ranking in future versions of our paper. This will help situate our research within the broader academic discourse and offer readers a more thorough understanding of the field.
>
> > **Q6:** From Equation (5), the response of the students belongs to 0 or 1, which means the student's responses studied in this paper are either right or wrong, what if the rating of the student responses has more choices (like the values in the range of [0,1])? Does the CCAT solution still work in this scenario?
>
> Indeed, changing the range of question values implies that the traditional IRT would no longer be applicable, and consequently, the CCAT algorithm cannot be directly used in its current form. However, we believe that our approach remains universal. Specifically, for any given evaluation model, we can still optimize the accuracy of ability ranking among collaborative students by enhancing the question selection process. Ultimately, collaborative student voting can be employed to rank the tested students effectively.
>
> > **Q7:** How to determine the parameter $T$ (testing round) in CCAT?
>
> In general, for research purposes in CAT problems, the parameter $T$ is often set to fixed values such as 5, 10, 15, or 20. These fixed values provide a standardized way to compare different methods and results. However, in practical application scenarios, the termination of T can be more dynamic. It can be based on specific indicator change amplitudes, such as the ability change value or ranking change value. The testing rounds can be terminated once these changes are less than a predefined threshold, ensuring that the testing adapts to the candidate's performance and achieves optimal efficiency.
>
> Reference:
>
> [1] Computerized adaptive testing: Theory and practice[M]. Dordrecht: Kluwer Academic, 2000.

---

> > ### Comment · Reviewer_gvzk · 2024-08-11
> >
> > Many thanks for the authors' rebuttals.  My questions and concerns have been well-addressed in the rebuttal, so I want to increase my recommendation score from 7 to 8.

---

> > > ### Author Response · Authors · 2024-08-12
> > >
> > > Thank you for your time and thoughtful evaluation. It's great to hear that all your questions and concerns have been successfully addressed.
> > >
> > > Your insights and suggestions are valuable to us, and we will include the discussed information in the future version.

---

### Official Review · Reviewer_cZqj · 2024-07-11

**Soundness:** 4
**Presentation:** 3
**Contribution:** 3
**Rating:** 7
**Confidence:** 5

**Summary:**

This paper addresses a real-world problem in AI education: improving the accuracy of student rankings by selecting different questions during the exam process. It proposes a question selection method based on collaborative students and provides theoretical guarantees. The experimental results have demonstrated the effectiveness of its method in ranking consistency.

**Strengths:**

1. To my knowledge, the perspective of this paper is novel. It starts from a real exam scenario and defines the CAT problem as a ranking problem, which has not been solved in previous CAT research. This means that based on this work, numerous ranking methods may be incorporated into CAT problems.

2. The paper demonstrates the method's superior ranking accuracy through experiments on real-world datasets. The approach exhibits general applicability across CAT systems estimated by IRT or GD.

3. The logic of this paper is clear, and the supplement materials include all theoretical proofs and several additional experiments, making this paper easy to follow.

**Weaknesses:**

1. Can you clarify the detail of \theta^T_c? Furthermore, since the true abilities of "collaborative students" are known, why use the abilities of collaborative students at T-moment instead of their true abilities.

2. The method proposed in the paper seems to have value primarily for educational research. While this is an important domain, the paper could benefit from discussing the potential applicability in other fields or providing more insights into how the proposed approach could be generalized beyond the education sector.

3. Ranking is a common issue in recommendation systems. After defining CAT tasks as ranking problems in this paper, what are the similarities and differences between CAT and recommendation tasks?

**Questions:**

See Weaknesses part

---

> ### Author Rebuttal · Authors · 2024-08-05
>
> Thank you for your valuable feedback!  Regarding the questions you raised, we have carefully considered each point and have made following responses:
>
> > **Q1:** Can you clarify the detail of $\theta^T_c$? Furthermore, since the true abilities of "collaborative students" are known, why use the abilities of collaborative students at T-moment instead of their true abilities.
>
> The parameter $\theta^T_c$ represents the ability assessment results obtained by collaborative students who have answered the same T questions as the students being tested. This approach ensures that the collaborative students' ability assessments are based on a comparable set of questions, providing a more accurate basis for scoring the tested students.
>
> Using the true abilities of collaborative students ($\theta^*_c$) for each student would mean that the collaborative students would lose their sensitivity to the question selection process. This would hinder the optimization of ranking through question selection, as the adaptive nature of the test would be compromised.
>
> > **Q2:** The method proposed in the paper seems to have value primarily for educational research. While this is an important domain, the paper could benefit from discussing the potential applicability in other fields or providing more insights into how the proposed approach could be generalized beyond the education sector.
>
> Our method performs particularly well when the number of CAT test questions is small, addressing the cold start problem inherent in CAT. This characteristic suggests that our research may be applicable to cold start problems in other fields as well. For instance, our approach could be valuable in personalized recommendation systems, where limited initial data is a common challenge. Additionally, it could be useful in any domain that requires adaptive testing or assessment with sparse initial data.
>
> > **Q3:** Ranking is a common issue in recommendation systems. After defining CAT tasks as ranking problems in this paper, what are the similarities and differences between CAT and recommendation tasks?
>
> CF and CCAT share some underlying principles, but they are fundamentally different in their applications and objectives. CF is a recommendation technique that estimates item preferences based on previous users' behaviors, while CCAT adapts the test items based on the examinee's ability level, which is estimated dynamically during the test.
>
> Moreover,CF aim to suggest items based on user preferences (**ranking items**), whereas CAT necessitating precise estimation of students' abilities with minimal interaction (**ranking users**). In CF, the item ranking of a user is fixed, but in CAT, selected questions affect the ability evaluation through the answers, impacting the ranking. This variability makes CAT tasks more challenging than typical CF problems.

---

> > ### Comment · Reviewer_cZqj · 2024-08-12
> >
> > Thank you for your response. After considering your feedback, I have decided to maintain my score.

---

> > > ### Author Response · Authors · 2024-08-14
> > >
> > > Thank you for your time and thoughtful evaluation. Your insights have been invaluable to us and will certainly help in refining our research.

---

### Official Review · Reviewer_M6H4 · 2024-07-12

**Soundness:** 2
**Presentation:** 1
**Contribution:** 2
**Rating:** 5
**Confidence:** 2

**Summary:**

The paper proposes an algorithm for performing computerized adaptive testing that handles and accounts for student rankings in the item recommendaiton.

**Strengths:**

- I like the idea of incorporating additional information in the collaborative filtering approach (while I don't quite understand why you need to intermittently rank students during CAT)

**Weaknesses:**

My main critique of the paper is the work's motivation for connecting ranking and CAT: Ranking seems like a completely separate task from CAT, and would happen after CAT. The paper's introduction would benefit from a concrete example where online updates in collaborating filtering is important when administering tests. It seems like there would be issues during testing such as handling the nonstationarity of question difficulties and forcing students to start and complete every question at the same time before proceeding onto the next question. It's not clear why you need intermediate ranks amongst students as they are getting tested.

Looking at Alg 1, it feels like the work should use/estimate student rankings not for the sake of estimation/collaborative filtering, but rather selecting questions that best *differentiate* students -- so distinguish their abilities to further diagnose students. But I'm curious what the authors originally had in mind with connecting online ranking estimations with CAT, because currently... I don't quite see how the algorithm assumptions can hold in the real world (see paragraph above).

**Questions:**

- Could the authors describe how their work differs from collaborative filtering approaches where estimation of item difficulty and user ability can be done through previous users and their attempted items?

- It seems strange to me that the collaborative records used for testing are being simultaneously updated at every step t (rf. Algorithm 1). Why do we need to update the student ranks online?

- Theorem 1 seems more like a statement about the collaborative filtering approach in estimating item difficulty and user ability, and is not really about estimating the ranking. Could the authors cleave the effects of items and users on the IRT estimation from the ranking estimation in their Theorem?

- While Figure 3 points to areas of improvements, I also noticed areas where their method does worse: e.g., on the second row of the NIPS-EDU. Could the aggregate difference be reported instead of the heat map visualization?

**Limitations:**

The authors do not provide a separate Limitations section in their paper. While the authors state they provided the limitations in Appendix D and experiments, these seem to be more like "findings" from empirical observations than limitations of their work/framing/approach.

---

> ### Author Rebuttal · Authors · 2024-08-05
>
> Thank you for your feedback on our manuscript. We sincerely appreciate your effort in evaluating our work. Below, we address each of your comments in detail:
>
> > **Q1**: My main critique of the paper is the work's motivation for connecting ranking and CAT: Ranking seems like a completely separate task from CAT, and would happen after CAT.
>
> You are right. Ranking typically occurs after the CAT process, but it is inherently interconnected with CAT and is also influenced by CAT . Existing CAT methods focus solely on the accuracy of abilities (Figure 4), overlooking the importance of ranking in CAT (Figure 1). This oversight can lead to potential biases and inequities in selection tests. Our research aims to address this gap by enhancing the testing process.
>
> > **Q2:** Could the authors describe how their work differs from collaborative filtering approaches?
>
> CF and CCAT share some underlying principles, but they are fundamentally different in their applications and objectives. CF is a recommendation technique that estimates item preferences based on previous users' behaviors, while CCAT adapts the test items based on the examinee's ability level, which is estimated dynamically during the test.
>
> Moreover, CF aim to suggest items based on user preferences (**ranking items**), whereas CAT necessitating precise estimation of students' abilities with minimal interaction (**ranking users**). In CF, the item ranking of a user is fixed, but in CAT, selected questions affect the ability evaluation through the answers, impacting the ranking. This variability makes CAT tasks more challenging than typical CF problems.
>
> > **Q3:** It seems like there would be issues during testing such as handling the nonstationarity of question difficulties and forcing students to start and complete every question at the same time.
>
> We would like to clarify that in traditional CAT models, the difficulty of questions is pretrained before testing and is fixed throughout the testing process. This assumption is based on the fact that the difficulty of a question is determined by its inherent attributes [1].
>
> Meanwhile, students do not need to wait for other students before proceeding onto the next one. Each student, upon completing a question, will compare their performance with a group of "collaborative students", which are defined as students who have already answered questions in question bank in Definition 1. Generally speaking, in real CAT systems such as GRE, a group of students will take tests in advance to generate test records [2], which indicates that our define and hypothesis is reasonable.
>
> > **Q4:** Why are intermediate ranks needed amongst students as they are getting tested?
>
> We do not dynamically update students' rankings during the CAT process. We only update students' abilities after each question (like other CAT) and use the comparison results between the tested students and collaborative students for rankings after testing.
>
> > **Q5:** Why are collaborative records used for testing being simultaneously updated at every step t (ref. Algorithm 1)?
>
> Collaborative records are not updated every round. Instead, we extract useful records from the collaborative records corresponding to the questions that the current student has answered.
>
> > **Q6:** Could the authors cleave the effects of items and users on the IRT estimation from the ranking estimation in their Theorem?
>
> Yes. Collaborative students are only a prerequisite of Theorem 1 rather than what it focuses on. Theorem 1 can be understood as a process where collaborative students vote for Student A and Student B (A is better than B). It claims that as long as there are enough collaborative students, A's votes will surpass B's votes. This explains why we utilize collaborative students to vote for ranking the students being tested.
>
> > **Q7**: Could the aggregate difference be reported instead of the heat map visualization?
>
> Yes. We originally hoped to visually demonstrate the advantages of CCAT's results compared to IRT's results through a heatmap. Below is the aggregate difference representing the average improvement of each student in 20 steps using CCAT compared to IRT (positive value means improve and negative value means decline):
>
> | Aggregate Difference | Average | 1     | 2     | 3    | 4     | 5    | 6    | 7     | 8    | 9    | 10   |
> | -------------------- | ------- | ----- | ----- | ---- | ----- | ---- | ---- | ----- | ---- | ---- | ---- |
> | NIPS-EDU             | 0.18$\uparrow$    | -0.15 | 0.18  | 0.22 | 0.19  | 0.13 | 0.24 | 0.22  | 0.49 | 0.06 | 0.16 |
> | JUNYI                | 0.004$\uparrow$   | -0.08 | -0.09 | 0.11 | -0.12 | 0.26 | 0.09 | -0.21 | 0.00 | 0.07 | 0.01 |
>
> Our method may not achieve better ranking relationships among all students, but from the table and various experimental results, our method can indeed improve the overall ranking consistency of students.
>
> > **Q8:** While the authors state they provided the limitations in Appendix D and experiments, these seem to be more like "findings" from empirical observations than limitations of their work.
>
> Sorry for the confusion. As mentioned in the Experiment section, we have stated limitations in line 262 that CCAT may not perform as well on long test sequences as methods that directly optimize capabilities. To make it clearer, we will separate statements of limitations into a new section in the new version of this paper.
>
> Reference:
>
> [1] Wainer H, Dorans N J, Flaugher R, et al. Computerized adaptive testing: A primer[M]. Routledge, 2000.
>
> [2] Computerized adaptive testing: Theory and practice[M]. Dordrecht: Kluwer Academic, 2000.

---

> > ### Comment · Reviewer_M6H4 · 2024-08-10
> >
> > Thanks for your response and clarification -- I see, so you're using the previous traces of other participants to do selection, instead of just the current participant's ability and the difficulties from your item bank.
> >
> > I've raised my score. I am curious, when authors say that that their algorithm shows "significant improvement", have you run a statistical sig test to verify this? I think it's generally good practice to state the test (at least in a footnote) and mark with the appropriate sig values when using "significant" as a descriptive.
> >
> > Thanks!

---

> ### Author Response · Authors · 2024-08-11
>
> Thank you for your positive feedback. We greatly appreciate your recognition of our efforts to address your concerns.
>
> Regarding your follow-up question, we did conduct a statistical significance test to verify the improvements reported in our algorithm. As shown in Tables 4 and 5 of Appendix D, we performed the tests and found significant results at the p < 0.05 level. However, we apologize for not explicitly mentioning this in the main text of our paper. We will ensure to include this information in the revised version of the main text, with the appropriate significance values clearly marked.
>
> If you have any other questions, please feel free to ask.

---

### Official Review · Reviewer_piWg · 2024-07-12

**Soundness:** 3
**Presentation:** 3
**Contribution:** 3
**Rating:** 8
**Confidence:** 5

**Summary:**

This paper proposes a new perspective on Computerized Adaptive Testing (CAT) by framing it as a task of ranking students. The authors define CAT as a ranking problem and present a feasible optimization algorithm to address this. Extensive experimental results demonstrate that this method significantly improves the consistency of student ranking scores compared to the baseline system.

**Strengths:**

1. The paper explores a previously overlooked issue in CAT and human assessment domain—student ranking consistency. It redefines the CAT task within this context. The solution proposed is interesting and interpretable, aligning well with real educational scenarios.
2. In terms of technical implementation, this paper utilizes existing students as ranking anchors to enhance selection and evaluation methods. It provides a theoretical basis, demonstrating that the algorithm can reduce ranking consistency errors to an acceptable level.
3. Experiments on real-world datasets have shown that the proposed method improves ranking consistency by an average of 5% compared to baseline selection methods.

**Weaknesses:**

1. This paper has rich and convincing experiments, but I want to know why Table 1 only shows the results of BOBCAT in the Table 1(a). Furthermore, this paper does not explain why BOBCAT has such a significant difference in performance on this task.
2. In Table 1 (a), NCAT performs best on the NIPS-EDU dataset at T=5, but this result is not  discussed in this paper.
3. This paper mainly analyzes the question selection and estimation of the CCAT method. In fact, according to my understanding, the construction of collaborative students is long-term and complex in real educational scenarios. Is there a method to ensure the effective construction of collaborative students?

**Questions:**

See above.

**Limitations:**

The paper and its supplementary materials clearly illustrate its limitations. Due to the need for collaborative students to be stored, the algorithm’s time and space complexity should be more thoroughly discussed. Providing detailed analyses and potential optimization strategies could strengthen the paper.

---

> ### Author Rebuttal · Authors · 2024-08-05
>
> We appreciate your comments! To address your concerns, below we prudently justify the details of our proposed method and experiments.
>
> > **Q1:** This paper has rich and convincing experiments, but I want to know why Table 1 only shows the results of BOBCAT in the Table 1(a). Furthermore, this paper does not explain why BOBCAT has such a significant difference in performance on this task.
>
> As a bilevel optimization model, BOBCAT optimizes both the IRT model and the question selection strategy. This means it employs a separate GD model and does not require training an additional IRT model through the MCMC method. Consequently, only one result for BOBCAT is shown in Table 1(a). The significant difference in BOBCAT's performance can be attributed to its two-layer optimization approach, which leads to peak accuracy at specific times (as illustrated in Appendix Figure 4, ACC), but poor performance at other times. Additionally, training the IRT model can lead to instability in estimating real ability, ultimately resulting in poor performance in ranking metrics.
>
> > **Q2:** In Table 1 (a), NCAT performs best on the NIPS-EDU dataset at T=5, but this result is not discussed in this paper.
>
> The CAT ranking problem we are studying can essentially be seen as an degradation problem for accuracy. When optimizing for accuracy, we may randomly achieve higher or lower ranking consistency. As observed in Table 1, although NCAT performs well on the NIPS-EDU dataset with T=5, its performance on the JUNYI dataset is even worse than that of random selection. This indicates that the NCAT method may not be effective in consistently optimizing ranking problems. We will include a discussion of these observations in the revised version of our paper to provide a comprehensive analysis of NCAT's performance across different datasets.
>
> > **Q3:** This paper mainly analyzes the question selection and estimation of the CCAT method. In fact, according to my understanding, the construction of collaborative students is long-term and complex in real educational scenarios. Is there a method to ensure the effective construction of collaborative students?
>
> Currently, the selection and construction of collaborative students are relatively simple processes. Generally speaking, in real CAT systems such as GRE, a group of students will take tests in advance to generate test records [1]. If CAT is viewed as a long-term process, while CAT increases students' learning efficiency, it also leads to the sparsification of student data. As data becomes sparser, the effectiveness of CAT and IRT may be impacted. To date, only one study has addressed the bias in IRT data [2]. Therefore, more research is needed to ensure the long-term construction and maintenance of collaborative students.
>
> Reference:
>
> [1] Computerized adaptive testing: Theory and practice[M]. Dordrecht: Kluwer Academic, 2000.
>
> [2] Kwon S, Kim S, Lee S, et al. Addressing Selection Bias in Computerized Adaptive Testing: A User-Wise Aggregate Influence Function Approach[C]//Proceedings of the 32nd ACM International Conference on Information and Knowledge Management. 2023: 4674-4680.

---

> > ### Comment · Reviewer_piWg · 2024-08-08
> >
> > These responses address my questions and strengthen my score.

---

> > > ### Author Response · Authors · 2024-08-09
> > >
> > > Thank you for your time and thoughtful evaluation. It's great to hear that all your questions have been successfully addressed and your acknowledgment means a lot to us.

---

### Official Review · Reviewer_74zz · 2024-07-14

**Soundness:** 3
**Presentation:** 3
**Contribution:** 2
**Rating:** 6
**Confidence:** 4

**Summary:**

The study deals with CAT( Computer Adaptive Testing) via Collaborative Ranking.

**Strengths:**

The proposed CCAT algorithm demonstrates superior performance in ranking 259 consistency across two public datasets. Particularly, CCAT shows more significant improvement 260 when fewer questions are tested, outperforming other methods.

**Weaknesses:**

The study does not discuss in sufficient detail the following: limitations, generalization possibility and the application of the algorithm as part of instructional design.

**Questions:**

N/A

---

> ### Author Rebuttal · Authors · 2024-08-05
>
> > **Q1:** The study does not discuss in sufficient detail the following: limitations, generalization possibility and the application of the algorithm as part of instructional design.
>
> Thank you for your valuable feedback. We apologize that we have shown in the experimental and appendix sections, rather than divided it into paragraphs for description and we will supplement this section in future versions of our paper:
>
> Limitations:
>
> As mentioned in section Experiment,  we have stated limitations in line 262 in this version that CCAT may not perform as well on long test sequences as methods that directly optimize capabilities. To make it clearer, we will separate statements of limitations into a new section in the new version of this paper.
>
> Generalization Possibility:
>
> Our method performs particularly well when the number of CAT test questions is small. Additionally, since CAT is inherently a cold start problem, our research may be applicable to other fields facing similar challenges. For example, personalized recommendation systems, healthcare diagnostics, and user behavior prediction are areas where cold start problems are prevalent, and our approach could potentially be generalized to improve performance in these fields.
>
> Application as Part of Instructional Design:
>
> In instructional design, the CCAT algorithm can be integrated to create adaptive learning paths tailored to individual student abilities. By dynamically adjusting the difficulty and selection of questions based on ongoing assessments, educators can provide a more personalized and effective learning experience. However, careful consideration must be given to the practical implementation and potential limitations discussed above.

---

### Decision · Program_Chairs · 2024-09-25

**Decision:**

Accept (poster)

**Comment:**

The paper argues that the ranking among students in tests are more important than the accurate estimation of student abilities, and proposes an algorithm called CCAT to optimize the ranking consistency on students instead of minimizing estimation errors on student abilities like existing work. Experiment results on two datasets show that CCAT can achieve higher ranking consistency than baselines and the improvement is particularly bigger with fewer questions.

The proposed approach is considered novel and the improvement is considered significant by most reviewers.

The authors are encouraged to incorporate reviewers’ feedback and suggestions into the final version to further improve the paper.